# Novel Toxicodynamic Model of Subcutaneous Envenomation to Characterize Snake Venom Coagulopathies and Assess the Efficacy of Site-Directed Inorganic Antivenoms

**DOI:** 10.3390/ijms241813939

**Published:** 2023-09-11

**Authors:** Vance G. Nielsen

**Affiliations:** Department of Anesthesiology, The University of Arizona College of Medicine, Tucson, AZ 85724, USA; vnielsen@anesth.arizona.edu

**Keywords:** envenomation, rabbit, coagulopathy, snake venom, ruthenium, platelets, plasma, thrombelastography

## Abstract

Venomous snake bite adversely affects millions of people yearly, but few animal models allow for the determination of toxicodynamic timelines with hemotoxic venoms to characterize the onset and severity of coagulopathy or assess novel, site-directed antivenom strategies. Thus, the goals of this investigation were to create a rabbit model of subcutaneous envenomation to assess venom toxicodynamics and efficacy of ruthenium-based antivenom administration. New Zealand White rabbits were sedated with midazolam via the ear vein and had viscoelastic measurements of whole blood and/or plasmatic coagulation kinetics obtained from ear artery samples. Venoms derived from *Crotalus scutulatus scutulatus*, *Bothrops moojeni*, or *Calloselasma rhodostoma* were injected subcutaneously, and changes in coagulation were determined over three hours and compared to samples obtained prior to envenomation. Other rabbits had ruthenium-based antivenoms injected five minutes after venom injection. Viscoelastic analyses demonstrated diverse toxicodynamic patterns of coagulopathy consistent with the molecular composition of the proteomes of the venoms tested. The antivenoms tested attenuated venom-mediated coagulopathy. A novel rabbit model can be used to characterize the onset and severity of envenomation by diverse proteomes and to assess site-directed antivenoms. Future investigation is planned involving other medically important venoms and antivenom development.

## 1. Introduction

Venomous snake bite is responsible for severe morbidity and mortality worldwide [1]. While antivenoms are available, they can be very expensive, injuries may still be severe/permanent, and antivenoms are not available for all venoms. Symptoms are caused by a myriad of enzymes, peptides, and other small molecular weight compounds that are contained in the snake venom, with metalloproteinases (SVMP), serine proteases (SVSP), and phospholipase A_2_ (PLA_2_) responsible for a great deal of loss of coagulation function, tissue damage, and paralysis [2,3,4,5,6,7,8,9,10,11,12,13]. In order to design and test novel antivenom strategies, a reliable preclinical animal model is needed that closely resembles humans, especially regarding the coagulation system. Rabbits could provide such a model, as their thrombelastographic profile and scanning electron micrograph images of whole blood and plasmatic thrombus formation are very similar to that of humans [14]. Initial studies with intravenous injection of the venom of *Crotalus atrox* demonstrated degradation of coagulation in a sedated rabbit with thrombelastography [15]. Further, in vitro exposure of this venom to tricarbonyldichlororuthenium (II) dimer (carbon monoxide releasing molecule 2, CORM-2) attenuated the effects of the venom in this rabbit model [16]. Thus, the effects of systemic envenomation in a minimally sedated rabbit model and attenuation of consequent venom-mediated coagulopathy by an inorganic antivenom were first documented [16].

However, most venomous snake bites do not involve the immediate release of most of the venom into circulation but instead are released via the lymphatic system [17,18,19], with some venous adsorption [19]. Thus, after physical disruption (via fangs) and venom enzymatic action, venom transport to the circulation would be expected to depend on the lymphatic flow [17,18]. As lymphatic flow is decreased by inhalational general anesthetics and likely decreased some by the immobility of intravenous anesthesia [20], a minimally sedated and humane animal model that involved no physical restraints following envenomation would more closely approximate the clinical situation of human envenomation. Other factors that may affect the toxicodynamic effects on coagulation following envenomation include venom injection into the same region and depth in the rabbit, as it has predictable regional lymphatic system anatomy [21]. Further, some venoms, such as that of *Bothrops asper*, can decrease lymphatic flow via the action of a myotoxic PLA_2_ [22]. Finally, in two studies, subcutaneous envenomation in rabbits to assess changes in coagulation with standard laboratory methods [23,24], despite being approved by their institutions, involved animals that were subjected to multiple envenomations without sedation or analgesia. A 50% mortality rate was observed in one, with severe tissue damage noted in the other, and animal vital signs were not recorded in both [23,24]. This manner of investigation is not acceptable; thus, the first goal of this study was to establish a minimally sedated, monitored, rabbit model of subcutaneous envenomation, as displayed in Figure 1A.

The selection of venoms in this work was based on the proteome, previous in vitro characterization of the coagulopathy observed via thrombelastography, and the likelihood that the hemotoxic enzymes of the venom would be inhibited by ruthenium-based, inorganic antivenoms [25,26]. The first venom, derived from *Crotalus scutulatus scutulatus* (type B) (Figure 1B), is hemotoxic and primarily fibrinogenolytic; a concentration of 250 ng/mL of venom diminishes the velocity of thrombus formation and final clot strength in human plasma by 75% [25]. This venom is Janus-like in its action on human platelets, as a concentration of 300 µg/mL decreases epinephrine-induced platelet aggregation by 25% [27], while a concentration of 40 mg/mL causes complete platelet aggregation [28]. Thus, given the multiple orders of magnitude of effect by concentration [25,27,28], it would be predicted that the primary effect of *C. s. scutulatus* would be a decrease in plasmatic coagulation in vivo. The second venom, derived from *Bothrops moojeni* (Figure 1C), is hemotoxic and possesses a prothrombotic proteome that would be expected to activate and consume platelets and plasmatic coagulation proteins [29]. Lastly, the venom obtained from *Calloselasma rhodostoma* (Figure 1D) is hemotoxic and contains a variety of procoagulant serine proteases, one in particular (ancrod) that was used medicinally for defibrinogenation, although consumption of both platelets and plasmatic coagulation proteins are observed [30]. Given the aforementioned, my working hypotheses were that: (1) the unique interactions of venom with the subcutaneous space would significantly affect systemic coagulation; (2) the novel Ru-based antivenom would neutralize the coagulopathy associated with venom administration. Therefore, the first goal of the present study was to create a novel rabbit model to characterize toxicodynamic profiles of the coagulopathies caused by these diverse venoms. Lastly, the second goal was to administer ruthenium-based antivenoms demonstrated to abrogate the anticoagulant [25] and procoagulant [26] effects of these venoms in vitro into the venom injection site, with the goal of diminishing coagulopathy. The composition of these antivenoms is displayed in Figure 2.

## 2. Results

### 2.1. Effects of C. s. Scutulatus Envenomation on Whole Blood and Plasmatic Coagulation and Efficacy of Antivenom

Rabbits were initially dosed with 1.4 mg/kg [31], but no effect was seen in whole blood or plasmatic coagulation. Subsequent doses of 2.8, 5.6, and 11.2 mg/kg were assessed, with the 11.2 mg/kg dose finally resulting in loss of plasmatic coagulation. A total of five rabbits were analyzed with this dose. A general observation was that the rabbits exhibited no signs of distress or behaviors of pain at the venom injection site. Further, there was no significant change in either heart rate or arterial blood oxygen saturation (SpO_2_) throughout experimentation. The results of the vital signs recorded are displayed in Table 1. It should also be noted that all raw data that were used to generate this table and all other subsequently presented tables and figures with more than n = 1 are presented in Appendix A.

The assessment of coagulation was performed with elastic modulus-based parameters defined as time to maximum rate of thrombus generation (TMRTG), which is the time interval (minutes or seconds) observed prior to maximum speed of clot growth; maximum rate of thrombus generation (MRTG), which is the maximum velocity of clot growth observed (dynes/cm^2^/second); and total thrombus generation (TTG, dynes/cm^2^), the final viscoelastic resistance observed after clot formation. Data were collected for 30 min. These variables, as measured from a whole blood and plasma sample obtained from a rabbit under baseline conditions, are displayed in Figure 3 with corresponding coagulation velocity curves. Also, the corresponding whole blood and plasma thrombelastic data observed on the computer screen as thrombi are formed in the device are presented on the right side of Figure 3. Lastly, in the series of experiments involving *C. scutulatus scutulatus* envenomation, the contribution of platelets to overall clot strength was determined with the following equation: Platelet mediated strength (%) = ((TTG of whole blood − TTG of plasma)/TTG of whole blood) × 100%.

With regard to coagulation, this dose of *C. s. scutulatus* venom had no significant effect on whole blood coagulation, as presented in Figure 4, left panel. In contrast, plasmatic coagulation had a significant decrease in MRTG values two and three hours after envenomation compared to baseline values, as noted in Figure 4, right panel. Further, a decrease in TTG values throughout experimentation after envenomation was also observed. Also of interest, there was no change in the contribution of platelet-mediated clot strength, as displayed in Figure 5. These data are characteristic of the effects of a primarily fibrinogenolytic venom with minimal effects on platelets [25,27,28].

The next phase of experimentation was to determine a dose of ruthenium-based antivenom to inject into the venom injection site. Given that the amount of venom required to cause this significant decrease in plasmatic coagulation was 8-fold greater than originally anticipated, it was decided to inject a large dose of CORM-2, 10 mg/kg, delivered as 10 mg/mL PBS. This single agent very effectively abrogated the anticoagulant effects of *C. s. scutulatus* venom in vitro [25].

As this venom only affected plasmatic coagulation, only the results from data collected from plasma samples are displayed in Figure 6. As can be observed in the left panel of Figure 6, the first experiment involving the injection of 10 mg/kg CORM-2 appeared to preserve the velocity of clot formation and strength until the third hour after venom injection. Subsequently, in the next experiment, 20 mg/kg of CORM-2 was administered, and it seemed that both the velocity of clot formation and clot strength were minimally affected for the three hours after venom injection. No remarkable changes in clinical signs were noted after injection of CORM-2 at these doses.

Encouraged by these results, it was planned to proceed with a new series of experiments with rabbits administered the previously mentioned dose of venom without or with antivenom administered. However, the preliminary studies used nearly all the venom purchased as the dose required was 8-fold greater than anticipated. The author was disappointed to learn that the source of the venom, the NNTRC in Texas, did not have sufficient venom to provide for these anticipated studies and was without a snake to collect more venom. Other sources of venom could not be identified by the NNTRC, so the author is unable to provide further information concerning *C. s. scutulatus*, type B, envenomation with this novel rabbit model at this time. Nevertheless, the author was able to learn from the aforementioned data and experiences to subsequently proceed with investigations with the subsequently presented venoms, which are in plentiful supply.

### 2.2. Effects of B. moojeni Envenomation on Whole Blood Coagulation and Efficacy of Antivenom

The first rabbit of this series was dosed with 3.0 mg/kg [32]. While the first blood sample collected before envenomation was separated into whole blood and plasma without any problem, the sample collected one hour after envenomation was observed to have the citrate anticoagulated whole blood sample containing some clotted material. Further, the plasma retrieved from the centrifuged sample was also solidified despite anticoagulation. It was apparent that this venom contained calcium-independent, procoagulant enzymes that were rapidly acting at this dose. The calculation of the contribution of platelet-mediated and plasma-mediated clot strength was not possible under these circumstances. To deal this this issue, it was decided to rapidly collect whole blood and place it into thrombelastographic cups with immediate activation with tissue factor as noted in the previously described methods. The time of collection to the time of onset of analysis was routinely under one minute, which should have outcompeted the procoagulant venom enzymes and allowed an assessment of tissue factor-initiated coagulation with the remaining components of the rabbit’s blood.

The second and third rabbits were administered 1.5 mg/kg of *B. moojeni* venom, and they displayed a consistent and remarkable pattern of coagulopathy over the three hours of observation after envenomation. Thus, this was the dose subsequently used for the remainder of experiments with this venom.

With regard to the composition and dose of antivenom chosen to be administered after envenomation, a combination of CORM-2 and RuCl_3_ was demonstrated to more effectively inhibit the procoagulant activity of the venom compared to either ruthenium-containing compound in isolation [26]. Further, as 20 mg/kg CORM-2 appeared to be potentially more effective than 10 mg/kg, the larger dose was selected. This larger dose would also be expected to compensate for any nonspecific binding of Ru-containing compounds with interstitial molecules that were not venom-related. Thus, the dose administered was CORM-2 10 mg/mL in a PBS containing 500 µM RuCl_3_ solution at a dose of 2 mL/kg. Lastly, after administration, the antivenom appeared to be adsorbed quickly in under 15 min.

With regard to the clinical state of the rabbits, as with the previous series of experiments, the rabbits exhibited no signs of distress or behaviors of pain at the venom injection site, a pattern that persisted either without or with the addition of antivenom injection. Further, there was no significant change in either heart rate or SpO_2_ throughout experimentation. The results of the vital signs recorded are displayed in Table 2 and Table 3.

As displayed in Figure 7, *B. moojeni* venom had little effect on coagulation one hour after injection. However, by two hours, envenomed rabbits demonstrated a significant loss of coagulation function, as evidenced by an increase in TMRTG and a decrease in both MRTG and TTG values. This loss of coagulation function became far more severe by the third hour in envenomed rabbits. In sharp contrast, rabbits injected with antivenom did not have any significant change in TMRTG values over the three hours post-venom injection. Further, while there was a significant decrease in MRTG and TTG values during the second and third hour following venom injection, rabbits administered antivenom had significantly less deterioration of coagulation function compared to animals not administered antivenom. Lastly, the interaction of time and antivenom administration was significant in the cases of TMRTG and TTG values, as seen in the top and bottom panels of Figure 7.

### 2.3. Effects of C. rhodostoma Envenomation on Whole Blood Coagulation and Efficacy of Antivenom

The first rabbit of this series was dosed with 3.0 mg/kg [33]. It should be noted that only whole blood samples were analyzed with this venom, as it was also procoagulant and likely to clot samples that were exposed to citrate anticoagulation. The pattern of coagulopathy with this rabbit and a second rabbit administered 3.0 mg/kg appeared somewhat similar to that observed with *B. moojeni* with 1.5 mg/kg. Subsequently, this was the dose of venom chosen for the remainder of this series of experiments with *C. rhodostoma* venom. The composition and dose of antivenom injected were the same as that used with *B. moojeni*, as the procoagulant activity of this venom was also more effectively inhibited by the two ruthenium-containing compounds in combination compared to either in isolation [26].

With regard to the clinical state of the rabbits, as with the previous two series of experiments, the rabbits exhibited no signs of distress or behaviors of pain at the venom injection site or after of antivenom injection. One rabbit in the group, administered venom alone, could not have HR or SpO_2_ analyzed with the rest of the data set secondary to monitor malfunction and loss of two time points until another monitor could be obtained from the veterinary staff. With the exception of the baseline measurement of heart rate, there was no significant change in either heart rate or SpO_2_ throughout experimentation within or between the two groups. The results of the vital signs recorded are displayed in Table 4 and Table 5.

As displayed in Figure 8, *C. rhodostoma* venom degraded coagulation throughout the observation period. A significant decrease in MRTG and TTG values was observed compared to baseline values at all three hours in rabbits injected with venom alone. A significant increase in TMRTG values was only observed at the three-hour time point in animals administered venom without antivenom administration. By the third hour, coagulation function was markedly diminished in this group, very similar in magnitude to the decrease seen with *B. moojeni* envenomation displayed in Figure 7. In sharp contrast, rabbits injected with antivenom did not have any significant change in TMRTG values over the three hours post-venom injection, and TMRTG was significantly smaller in this group compared to the animals injected with venom alone. Further, while there was a significant decrease in MRTG and TTG values throughout the observation period following venom injection, rabbits administered antivenom had significantly less deterioration of coagulation function compared to animals not administered antivenom. Lastly, the interaction of time and antivenom administration was significant in the cases of TMRTG, MRTG, and TTG values, as seen in the panels of Figure 8.

## 3. Discussion

The present study successfully achieved its stated goals regarding the toxicodynamic characterization of diverse venoms with this minimally sedated rabbit model and assessment of the efficacy of ruthenium-based antivenoms. Both goals will be subsequently discussed in detail.

The toxicodynamic characterizations of the three venoms chosen were remarkable. First, in the case of *C. s. scutulatus* envenomation, a remarkable amount of venom (11.2 mg/kg) was needed to cause a significant decrease in plasmatic coagulation compared to the small concentration (250 ng/mL) required to compromise human plasmatic coagulation [25]. Possible mechanisms by which such large doses of venom were needed to cause changes in plasmatic coagulation include poor enzymatically mediated egress into the lymphatic system (given the minimal vascular trauma caused by a 25G needle) or perhaps remarkable redistribution and elimination from the circulation of the animal. The relative plateau in compromised plasmatic coagulation after injection of *C. s. scutulatus* venom supports the concept of redistribution of venom enzymes from target molecules (e.g., fibrinogen), a phenomenon observed when a bolus of *Crotalus atrox* (Western diamondback rattlesnake) venom was injected intravenously into the rabbit [15,16]. Lastly, the use of whole blood and plasma samples demonstrated that this venom primarily affects plasmatic coagulation, without significant changes in platelet-mediated coagulation noted. Regarding the second venom investigated, *B. moojeni* envenomation demonstrated a one-hour “pause” after injection, followed by a rapid degradation of whole blood coagulation over the subsequent two hours of observation. This is consistent with the concept that this venom interfered with the lymphatic flow, as observed with other *Bothrops* species [22]. The pattern of rapid destruction of whole blood coagulation by *B. moojeni* venom is remarkably different from that of *C. s. scutulatus* venom, and it is very likely that whatever rate redistribution or elimination of *B. moojeni* venom occurs in the rabbit is outcompeted by the rate of catalysis of the venom enzymes. Lastly, the toxicodynamic pattern of coagulopathy of *C. rhodostoma* venom was the most impressive as there was no “pause” in the onset of whole blood coagulopathy as seen with *B. moojeni* venom. *C. rhodostoma* venom at the dose administered relentlessly destroyed coagulation function over the three-hour observation period, demonstrating rapid entry into the lymphatic system and likely continuous consumption of both cellular and plasmatic elements of coagulation. In summary, this rabbit model allowed for the identification of the toxicodynamic “fingerprint” of these three venoms that are diverse in proteome and geographical origin.

Assessment of the efficacy of the ruthenium-based antivenoms was largely successful, with the limitation that there was insufficient *C. s. scutulatus* venom available to perform enough experiments to statistically compare groups administered venom only or venom followed by antivenom injection. However, preliminary experiments with this venom influenced the composition and dose of ruthenium compound containing antivenom tested to abrogate the coagulopathic effects of the other two venoms. Protection from prolongation of TMRTG values after injection of either *B. moojeni* or *C. rhodostoma* venom (Figure 6 and Figure 7) indicates that the antivenom prevented a critical loss of procoagulants that would prevent the normal onset of coagulation—a key function of hemostasis. Antivenom administration also decreased the velocity of the loss of MRTG and TTG values after *B. moojeni* or *C. rhodostoma* envenomation, with values several-fold greater at three hours post venom injection compared to animals without antivenom injection (Figure 6 and Figure 7). These patterns of protection are likely secondary to irreversible inhibition of key venom enzymes, with degradation of systemic coagulation caused by venom that either was not inhibited secondary to not being exposed to the antivenom within the injection site or perhaps by the venom gaining access to the lymphatic during the five minutes prior to antivenom injection. Thus, these data support the concept that this novel site-directed, ruthenium compound-based approach attenuated the venom-mediated degradation of whole blood systemic coagulation function.

While a small animal model was employed to conduct this investigation, the paradigm of antivenom treatment was not organism-focused but instead “bite site” focused. Put another way, treatment consisted of neutralizing the venom injected with direct antivenom application, not providing a circulating antivenom moiety with a long circulating half-life to inactivate venom enzymes as they enter the bloodstream. Given the potential nonspecific binding of the ruthenium radical formed during the release of carbon monoxide to other biomolecules in the subcutaneous space, a large dose of CORM-2 was justified and has previously been well tolerated when injected intravenously into rabbits [34]. Another dose of CORM-2, 20 mg/kg, was justified if 10 mg/kg was unsuccessful in diminishing venom activity, as up to 30 mg/kg of CORM-2 in a murine model of acute kidney injury was well tolerated and protected against injury [35]. CORM-2 doses that result in no toxicity in vivo vary a great deal, with values as small as 5 mg/kg to 50 mg/kg depending on the species [35,36,37,38,39,40,41,42,43,44,45,46,47,48,49,50,51,52]. As has been demonstrated with several venoms and venom enzymes, it is the ruthenium radical of CORM-2 that presumably binds to key amino acid residues, such as histidine, to inhibit activity [53]. Unfortunately, biomolecules such as albumin that are in the interstitial space contain such amino acids, which is what necessitates the administration of larger volumes and concentrations of ruthenium-based antivenoms to successfully neutralize antivenom activity despite nonspecific binding to other compounds. Also of interest, the dosing of the antivenom would be based on the amount of venom injected during the snake bite in larger organisms such as domestic animals and humans, so it is anticipated that a fixed dose of ruthenium-based antivenom would be required to neutralize a range of venom volumes. As an example, adult *Vipera russelli* have been documented to inject, on average, 63 mg and up to 147 mg of venom (after desiccation) per bite [54]; thus, it would need to be determined with the presented rabbit model or via clinical trials what fixed dosage of ruthenium-based antivenom would abrogate this amount of venom. Lastly, it should be noted that this site-directed approach is markedly different from traditional antibody-based antivenom therapy, as the Ru-based compounds most likely only act locally with molecular entities and do not enter the circulation in any meaningful amount. In sharp contrast, the use of traditional antivenoms administered systemically can only inactivate venom enzymes and other venom-containing compounds once they enter the circulation. In summary, while the administration of both venom and antivenom is based on kg of the rabbit, the rabbit is serving as a “bite site” to assess molecular interactions between venom and ruthenium-based antivenom.

This investigation has some limitations. First, it could be argued that five minutes seems to be a brief period prior to administering site-directed antivenom. While this is understandable, the consideration that the subcutaneous space of the rabbit is only a few millimeters thick and well-perfused drove the decision to choose this interval of time to delay antivenom treatment. Arguably, some venom will immediately enter the circulation secondary to the trauma of injection, and more venom will likely enter the lymphatic system at some unknown rate. Thus, no matter how effective the site-directed antivenom may be, the venom that has left the bite site will wreak havoc on the target cells and molecules in the circulation. This may be the scenario observed in the cases of envenomation by *B. moojeni* and *C. rhodostoma* when assessing changes in whole blood coagulation without or with antivenom administration in Figure 7 and Figure 8. Given that the rabbit model is an artificial construct to assess toxicodynamic change in coagulation and antivenom efficacy, these results are somewhat expected. Other limitations of this investigation include not assessing multiple doses of all venoms used in a variety of concentrations and compositions of ruthenium-based antivenom. Given that this work is a “proof-of-concept” work wherein the goals were limited and needless loss of animal life should be avoided, it is held that further experimentation is warranted, but with other venoms (e.g., *Crotalus atrox*, etc.) and antivenom doses in future works. Thus, despite these limitations, the present investigation achieved its goals. In conclusion, a novel animal model was used to characterize the toxicodynamics of distinct venom proteomes, varying from fibrinogenolytic anticoagulant and thrombin-like, prothrombotic venoms, identifying the components of the coagulation system that are damaged. Lastly, the translation of the in vitro findings that various ruthenium compounds containing antivenoms inhibited these various venoms [14,15] to the in vivo setting confirms that the molecular mechanisms of inhibition of snake venom enzymes [53] hold true in an acutely envenomed animal.

## 4. Materials and Methods

### 4.1. Chemicals and Venoms

Lyophilized venoms derived from *C. scutulatus scutulatus* (type B), *B. moojeni,* and *C. rhodostoma* were provided by the National Natural Toxins Research Center (NNTRC) located at Texas A&M University-Kingsville, Kingsville, TX, USA. The National Institutes of Health funds the NNTRC out of the Office of Research Infrastructure Programs. Venoms were dissolved into calcium-free phosphate-buffered saline (PBS, Millipore Sigma, Saint Louis, MO, USA) to a final 30 mg/mL concentration, aliquoted, and maintained at −80 °C. Dimethyl sulfoxide (DMSO), tricarbonyldichlororuthenium (II) dimer (CORM-2), and RuCl_3_ were obtained from Millipore Sigma (Saint Louis, MO, USA). Tissue factor for activating coagulation was obtained in the form of Pacific Hemostasis™ Prothrombin Time Reagent (Thermo Fisher Scientific, Pittsburgh, PA, USA). Calcium chloride (200 mM) was obtained from Haemonetics Inc. (Braintree, MA, USA).

### 4.2. Rabbit Model

Male New Zealand White rabbits (2–3 kg) were procured from Charles River Laboratories (San Diego, CA, USA) and housed within our animal facility and allowed food and water ad libitum for at least 1 week prior to experimentation. The Institutional Animal Care and Utilization Committee of the University of Arizona approved all procedures involving these rabbits (protocol #2022-0887). The protocol was conducted in accordance with all applicable federal and institutional policies, procedures, and regulations, including the PHS Policy on Humane Care and Use of Laboratory Animals, USDA regulations (9 CFR Parts 1, 2, 3), the Federal Animal Welfare Act (7 USC 2131 et. Seq.), the Guide for the Care and Use of Laboratory Animals, and all relevant institutional regulations and policies regarding animal care and use at the University of Arizona.

Rabbits were briefly restrained and had one ear closely clipped and cleaned with a 70% isopropyl alcohol pad. A 22 G plastic catheter was placed in the central ear artery, and another similar catheter was placed in the marginal ear vein; both catheters were connected to an end cap with a rubber diaphragm that allowed the withdrawal of blood samples and administration of medications. The animals were sedated intravenously with 1 mg/kg midazolam, with supplemental doses of 0.5–1 mg/kg provided during experimentation to maintain sedation. As displayed in Figure 1A, one toe of a forepaw was subsequently closely clipped, with a pulse oximeter probe placed to monitor heart rate (HR, beats/min) and % arterial oxygenation (SpO_2_) with a CMS60D-VET SPO2 Pulse Oximeter (CONTECTM, Qinhuangdao (Hebei), China). Heart rate (HR) and SpO_2_ were recorded at baseline and every 15 min thereafter until the end of the experiment.

An approximately 5 cm by 5 cm area of skin over either flank of the rabbit, midway between the lumbar spine and mid abdomen, was closely shaved and cleaned with a 70% isopropyl alcohol pad. A 1 cm circle was drawn with a felt tip marker in the center of this area, which served as the injection site for venom and antivenom as appropriate. After obtaining the baseline HR value, SpO_2_ value, and the initial blood sample, venom was injected subcutaneously in the middle of the circle through a 5/8 inch long, 25G needle (Thermo Fisher Scientific, Pittsburgh, PA, USA). The initial dose of each venom was based on a value obtained from mice lethal dose 50% studies (LD_50_), with the dose for rabbits beginning with approximately half of the LD_50_ dose. Thus, the initial subcutaneous doses for each venom were as follows: *C. scutulatus scutulatus* (1.4 mg/kg) [31], Figure 1B; *B. moojeni* (3.0 mg/kg) [32] Figure 1C, and *C. rhodostoma* (3.0 mg/kg) [33], Figure 1D. The dose of venom was increased or decreased until a consistent pattern of coagulopathy was observed; thereafter, this dose was used to characterize the coagulopathy and to test the efficacy of antivenom as subsequently described. Blood samples were collected from this time point onward every hour for 3 h. If the animal was to be administered antivenom, then this was performed 5 min after injection of the venom, again administered through a 25G needle(Thermo Fisher Scientific, Pittsburgh, PA, USA). Antivenom was composed of one of three doses and contents: (1) CORM-2 in PBS at a concentration of 10 mg/mL, administered at a dose of 1 mL/kg; and (2) CORM-2 10 mg/mL in a PBS containing 500 µM RuCl_3_ solution at a dose of 2 mL/kg. Antivenom solutions, as displayed in Figure 2, were made freshly just after the venom injection over a 3 min period prior to administration. Ten minutes after the venom injection, a 1.5 cm by 1.5 cm piece of a 4% lidocaine analgesic patch (Lidocaine Pain Relief Patch, Walgreens, Walgreen Company, Deerfield, IL, USA) was placed over the injection site to minimize discomfort for the remainder of the experiment. Lastly, after the last blood sample and vital sign assessments were obtained, the rabbits were euthanized with intravenous administration of 1 mL of pentobarbital/phenytoin (390/50 mg/mL).

### 4.3. Coagulation Monitoring

Blood was collected prior to envenomation, designated the baseline sample, and then every hour after envenomation for three hours. Blood collected during the experiments involving Mojave rattlesnake venom was placed into a sodium citrate-containing tube (2.7 mL blood; 9 parts blood to 1 part 0.105 M sodium citrate), with an aliquot removed for whole blood coagulation evaluation. The remaining blood was subjected to centrifugation at 3000× *g* for 15 min at room temperature, with plasma decanted and coagulation kinetics assessed as subsequently described. In experiments involving the Brazilian lancehead and Malayan pit viper, whole blood samples (1 mL) were quickly collected, with aliquots placed immediately into thrombelastographic (TEG) cups for analysis as subsequently presented. The rationale for this approach with the latter two venoms was that it was noted that blood could clot within the citrate-containing tubes within just a few minutes during preliminary experiments, indicative of venom enzymatic activity that was calcium-independent and would confound the coagulation assessment. Thus, to minimize this in vitro artifact, blood was rapidly placed into thrombelastographic cups with activation by tissue factor.

All sample mixtures were placed in a disposable cup in a computer-controlled thrombelastograph^®^ hemostasis system (Model 5000; Haemonetics Inc., Braintree, MA, USA) at 39 °C, the normal temperature of the NZW rabbit. The mixture used in the series of experiments involving Mojave rattlesnake venom was composed of 330 µL of whole blood or plasma, 10 µL of tissue factor (0.1% final concentration of Pacific Hemostasis™ Prothrombin Time Reagent, Thermo Fisher Scientific, Pittsburgh, PA, USA), and 20 µL of 200 mM CaCl_2_ (Haemonetics Inc.). In the series of experiments involving the Brazilian lancehead and Malayan pit viper, the sample mixture was 350 µL of whole blood and 10 µL of tissue factor. After mixing the samples by raising and lowering the cup to the level of the thrombelastographic disposable pin five times, the following elastic modulus-based parameters were determined: time to maximum rate of thrombus generation (TMRTG), which is the time interval (minutes or seconds) observed prior to maximum speed of clot growth; maximum rate of thrombus generation (MRTG), which is the maximum velocity of clot growth observed (dynes/cm^2^/second); and total thrombus generation (TTG, dynes/cm^2^), the final viscoelastic resistance observed after clot formation. Data were collected for 30 min. Lastly, the calculations concerning the contributions of platelets and plasma to clot strength were based on previous validated studies with rabbit human blood [55].

### 4.4. Statistical Analyses

Data are presented as mean ± SD. All experimental groups were represented by n = 5–7 different individuals, as this provided a statistical power > 0.8 with *p* < 0.05 using this methodology to assess differences in thrombelastographic parameters within and between groups. A commercially available statistical program was used for one-way or two-way, repeated measures analyses of variance (ANOVA) as appropriate to the dataset, followed by Holm–Sidak post hoc analyses (SigmaStat 3.1; Systat Software, Inc., San Jose, CA, USA). Graphics were generated with commercially available programs: Origen 2023, OrigenLab Corporation, Northampton, MA, USA; and CorelDRAW Home & Student, Alludo, Ottawa, ON, Canada). *p* < 0.05 was considered significant.

## 5. Conclusions

This investigation was successful in determining the toxicodynamics of the coagulopathies caused by various venom proteomes that are very different on the molecular level with a novel rabbit model. Further, translation of in vitro findings of inhibition of venom activity with novel combinations of site-directed ruthenium-based antivenoms was also achieved. Future investigations to characterize the coagulopathy caused by medically important snake venoms with diverse proteomes and determination of the efficacy of site-directed antivenoms are justified and planned. Ultimately, it is hoped that these preclinical studies can be translated to clinical studies involving both human beings and their animals envenomed by snakebite, finally resulting in a novel, site-directed therapy.

## Figures and Tables

**Figure 1 ijms-24-13939-f001:**
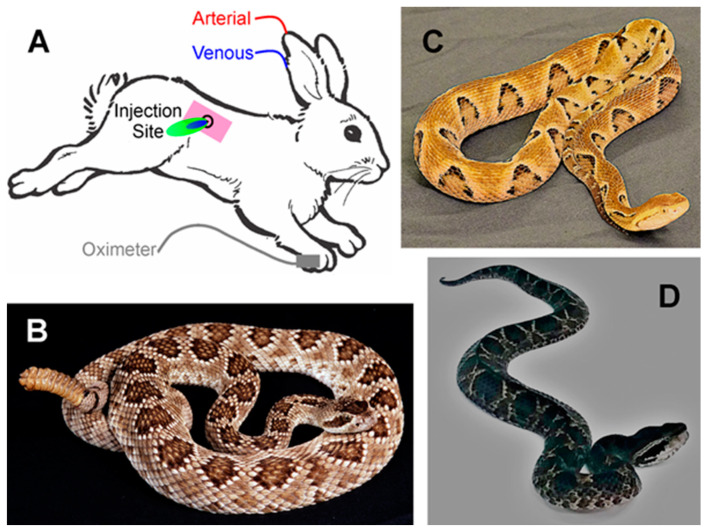
(**A**) Rabbit model of envenomation. Blue area indicates site of venom injection, with green indicating the antivenom injection. (**B**) The Mojave rattlesnake, *Crotalus scutulatus scutulatus*. (**C**) The Brazilian lancehead, *Bothrops moojeni*. (**D**) The Malayan pit viper, *Calloselasma rhodostoma*. The photographs of the snakes were kindly provided by the National Natural Toxins Research Center at Texas A&M University-Kingsville, Kingsville, TX, USA.

**Figure 2 ijms-24-13939-f002:**
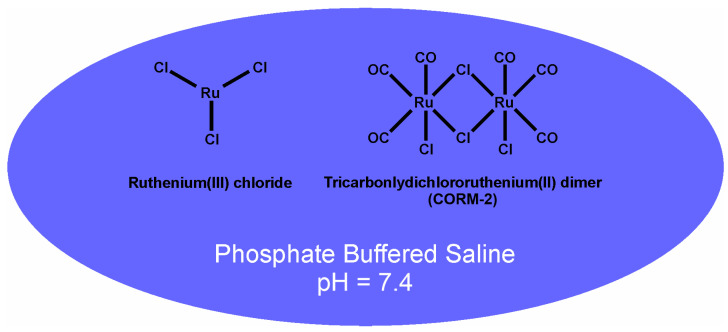
Ruthenium-based antivenoms. CORM-2 alone in phosphate-buffered saline (PBS) or a combination of RuCl_3_ with CORM-2 in PBS are the inorganic antivenoms utilized in the present investigation.

**Figure 3 ijms-24-13939-f003:**
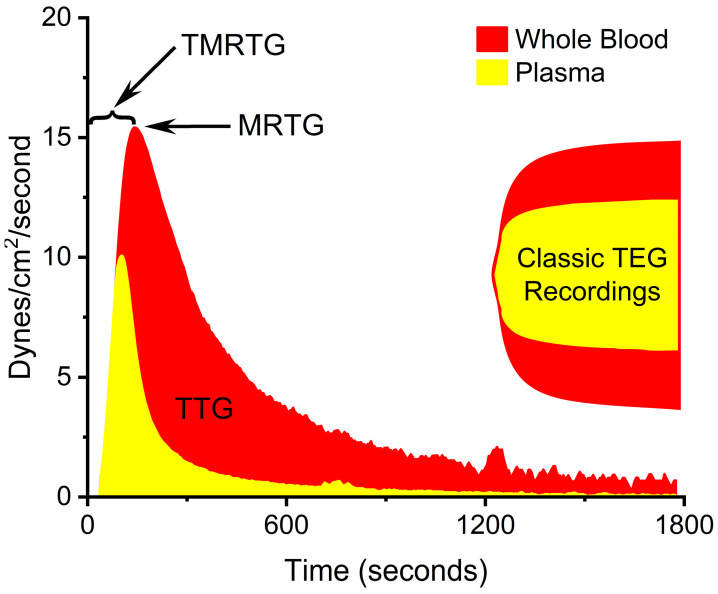
Thrombelastographic parameters displayed in clot growth velocity curves of rabbit whole blood and plasma. Typical classic, corresponding recordings of clot formation are displayed on the right side of the diagram. See the preceding text for definitions of the thrombelastographic variables.

**Figure 4 ijms-24-13939-f004:**
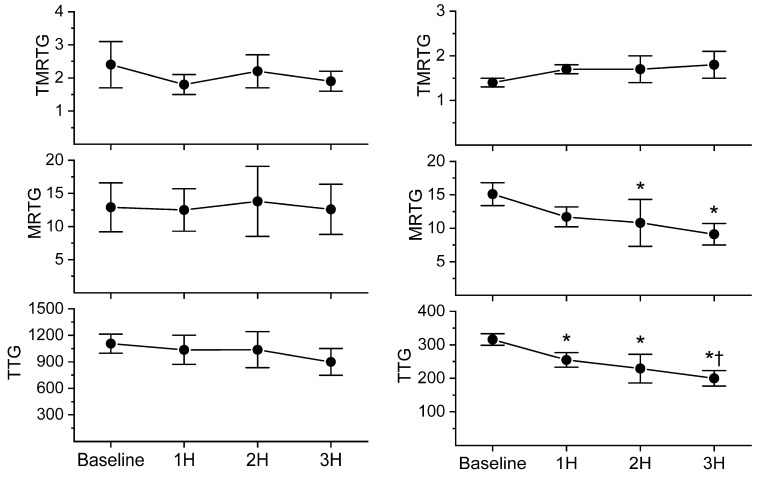
Comparison of whole blood and plasmatic coagulation after *C. s. scutulatus* envenomation. Left panel—There was no significant change in whole blood coagulation following envenomation with *C. s. scutulatus* venom. Right panel—*C. s. scutulatus* venom injection resulted in a significant decrease in MRTG (dynes/cm^2^/second) and TTG (dynes/cm^2^) without affecting TMRTG (minutes). Data presented as mean ± SD; 1H, 2H, and 3H are hours after venom injection; * *p* < 0.05 vs. baseline; † *p* < 0.05 vs. 1H.

**Figure 5 ijms-24-13939-f005:**
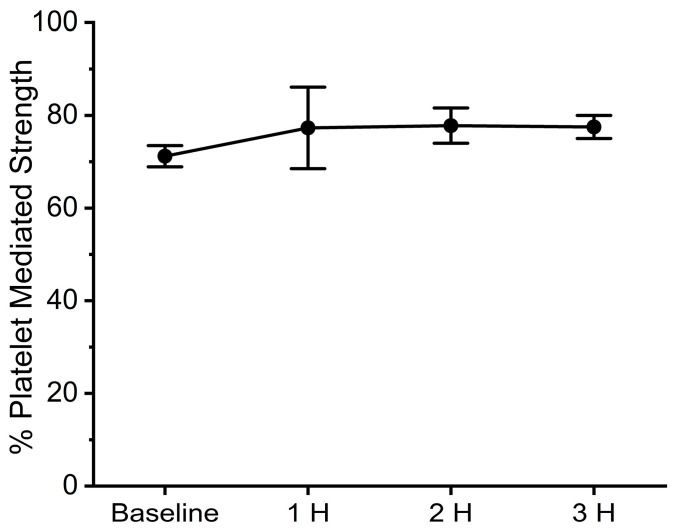
Contribution of platelets to clot strength after *C. s. scutulatus* envenomation. Data presented as mean ± SD.

**Figure 6 ijms-24-13939-f006:**
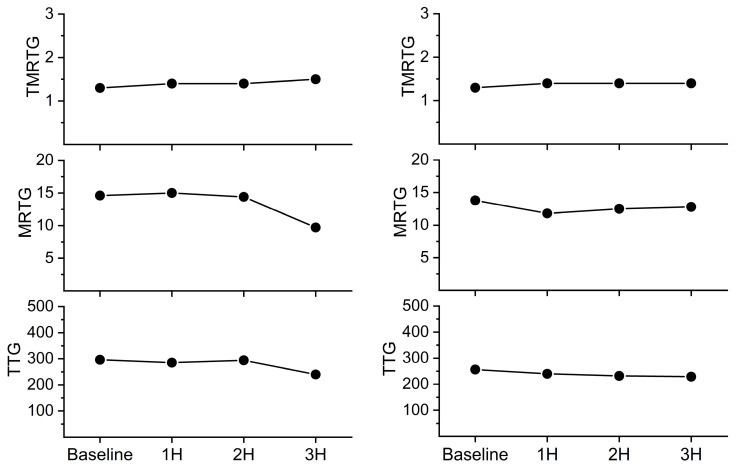
Effects of injection of CORM-2 into the *C. s. scutulatus* envenomation site on plasmatic coagulation kinetics. Left panel—10 mg/kg CORM-2 injected (n = 1); Right panel—20 mg/kg CORM-2 injected (n = 1).

**Figure 7 ijms-24-13939-f007:**
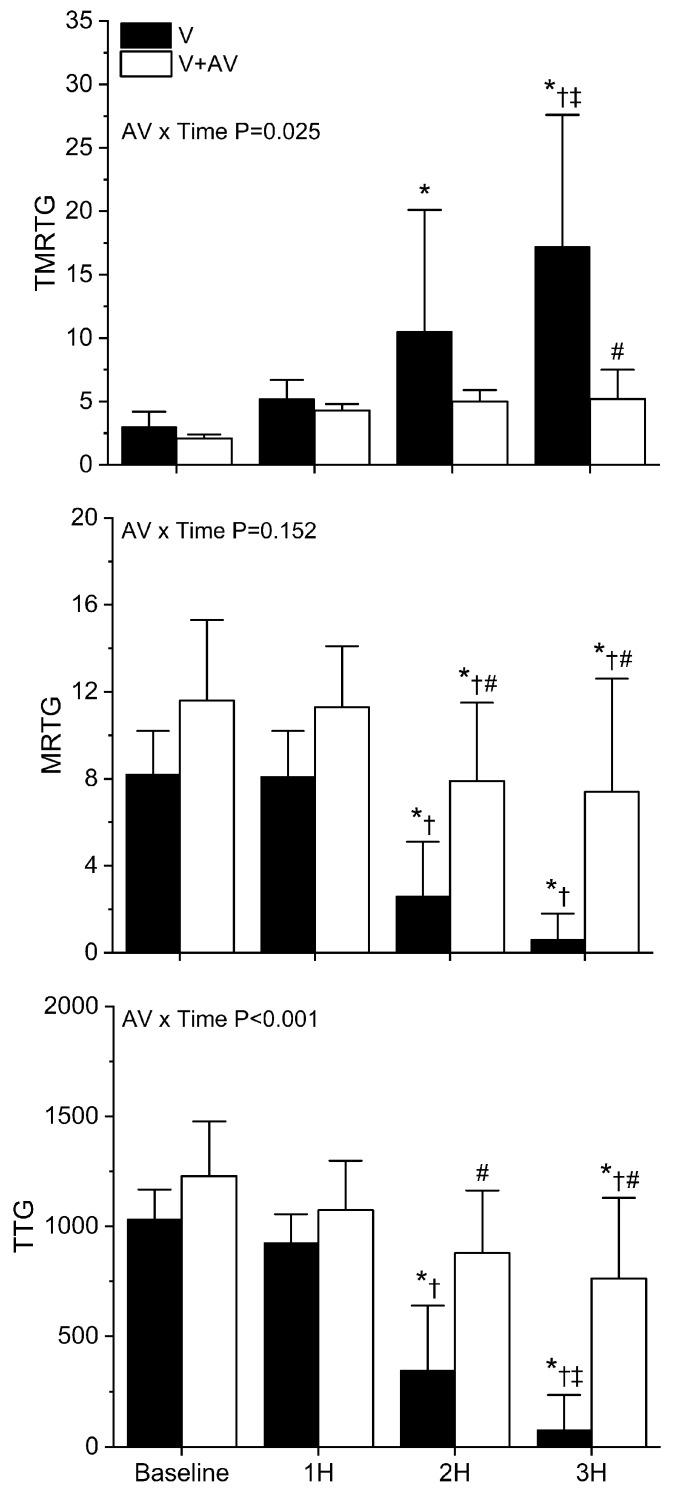
Effects of *B. moojeni* venom injection without or with antivenom treatment on whole blood coagulation. V (black bars) = rabbits injected with venom; V + A (white bars) = rabbits administered antivenom after venom injection. TMRTG (minutes); MRTG (dynes/cm^2^/second; TTG (dynes/cm^2^). Data presented as mean ± SD; 1H, 2H, and 3H are hours after venom injection; * *p* < 0.05 vs. baseline; † *p* < 0.05 vs. 1H; ‡ *p* < 0.05 vs. 2H; # *p* < 0.05 vs. V. AV × Time = result of two-way ANOVA to determine interaction of time with antivenom administration.

**Figure 8 ijms-24-13939-f008:**
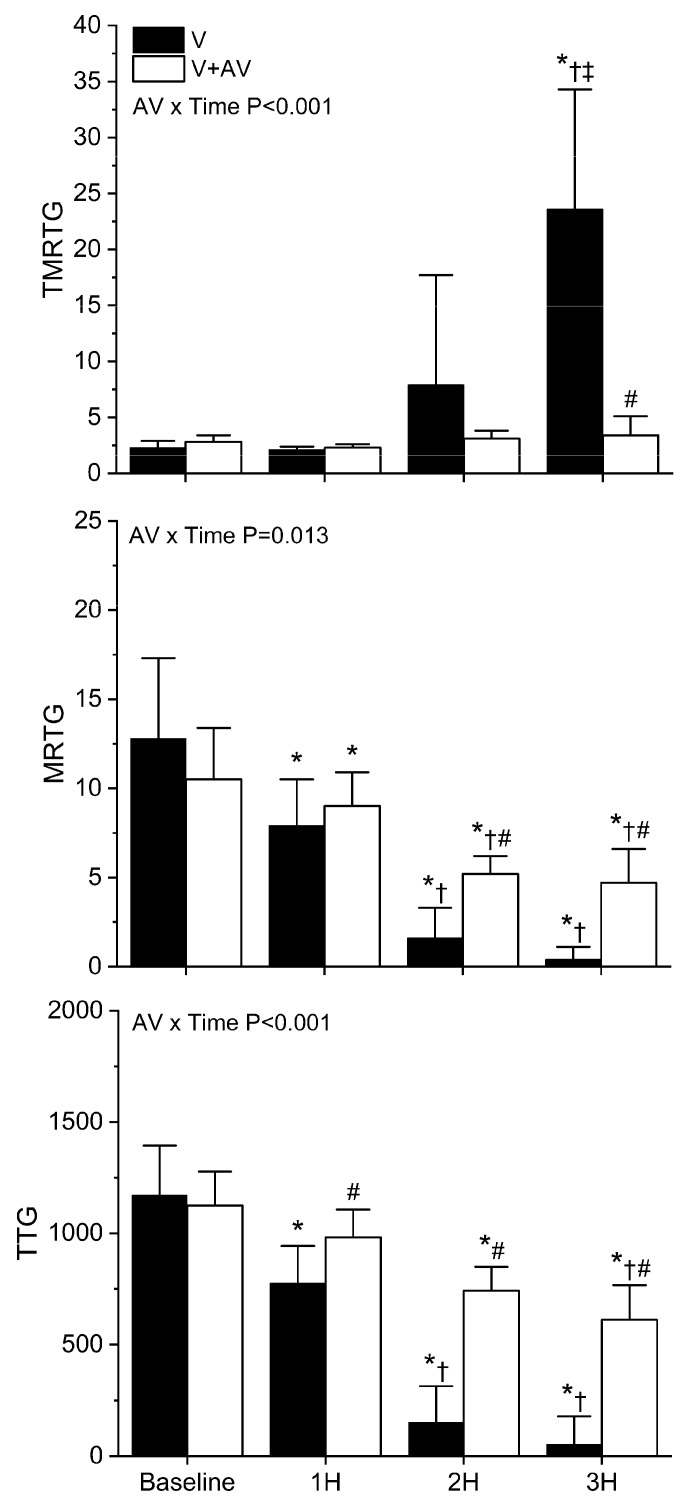
Effects of *C. rhodostoma* venom injection without or with antivenom treatment on whole blood coagulation. V (black bars) = rabbits injected with venom; V + A (white bars) = rabbits administered antivenom after venom injection. TMRTG (minutes); MRTG (dynes/cm^2^/second; TTG (dynes/cm^2^). Data presented as mean ± SD; 1H, 2H, and 3H are hours after venom injection; * *p* < 0.05 vs. baseline; † *p* < 0.05 vs. 1H; ‡ *p* < 0.05 vs. 2H; # *p* < 0.05 vs. V. AV × Time = result of two-way ANOVA to determine interaction of time with antivenom administration.

**Table 1 ijms-24-13939-t001:** HR and SpO_2_ during experimentation with *C. scutulatus scutulatus* venom.

Time	BSL	15	30	45	60	75	90	105	120	135	150	165	180
**HR**	192 ± 39	185 ± 33	183 ± 40	185 ± 33	176 ± 37	171 ± 12	168 ± 14	169 ± 8	188 ± 35	177 ± 19	172 ± 17	178 ± 13	168 ± 22
**SpO_2_**	98 ± 1	98 ± 1	98 ± 1	98 ± 2	96 ± 3	97 ± 2	98 ± 1	98 ± 1	98 ± 1	98 ± 1	97 ± 2	95 ± 2	98 ± 2

Time: BSL = baseline; the remaining numbers are time in minutes after venom injection. Data presented as mean ± SD. No significant changes noted by two-way analysis of variance (ANOVA).

**Table 2 ijms-24-13939-t002:** HR during experimentation with *B. moojeni* venom.

Time	BSL	15	30	45	60	75	90	105	120	135	150	165	180
**V**	218 ± 21	222 ± 18	208 ± 36	194 ± 35	214 ± 29	204 ± 34	196 ± 39	209 ± 29	214 ± 25	212 ± 25	218 ± 22	216 ± 38	219 ± 22
**A + V**	194 ± 41	222 ± 32	242 ± 15	203 ± 43	218 ± 26	226 ± 29	228 ± 28	217 ± 33	213 ± 29	221 ± 20	220 ± 14	221 ± 19	228 ± 18

Time: As in Table 1. V = venom injection; A + V = venom injection followed by antivenom injection. Data presented as mean ± SD. No significant changes noted by two-way ANOVA.

**Table 3 ijms-24-13939-t003:** SpO_2_ during experimentation with *B. moojeni* venom.

Time	BSL	15	30	45	60	75	90	105	120	135	150	165	180
**V**	98 ± 2	97 ± 1	96 ± 3	98 ± 2	96 ± 3	96 ± 3	98 ± 2	98 ± 1	99 ± 1	98 ± 2	98 ± 2	98 ± 2	98 ± 1
**A + V**	98 ± 2	96 ± 2	96 ± 3	97 ± 2	98 ± 1	98 ± 2	95 ± 3	97 ± 2	97 ± 2	97 ± 2	98 ± 2	97 ± 2	97 ± 2

Time: As in Table 1. V = venom injection; A + V = venom injection followed by antivenom injection. Data presented as mean ± SD. No significant changes noted by two-way ANOVA.

**Table 4 ijms-24-13939-t004:** HR during experimentation with *C. rhodostoma* venom.

Time	BSL	15	30	45	60	75	90	105	120	135	150	165	180
**V**	214 ± 20	224 ± 28	220 ± 32	216 ± 19	232 ± 21	231 ± 28	231 ± 15	236 ± 13	223 ± 22	227 ± 19	232 ± 23	242 ± 22	239 ± 19
**A + V**	239 ± 2 *	246 ± 6	237 ± 11	226 ± 16	216 ± 14	219 ± 9	218 ± 13	217 ± 9	218 ± 14	220 ± 13	219 ± 7	224 ± 9	223 ± 15

Time: As in Table 1. V = venom injection; A + V = venom injection followed by antivenom injection. Data presented as mean ± SD. * *p* < 0.05 vs. V using two-way ANOVA followed by Holm–Sidak post hoc analyses.

**Table 5 ijms-24-13939-t005:** SpO_2_ during experimentation with *C. rhodostoma* venom.

Time	BSL	15	30	45	60	75	90	105	120	135	150	165	180
**V**	97 ± 2	98 ± 2	98 ± 1	98 ± 1	97 ± 2	97 ± 2	98 ± 2	98 ± 2	98 ± 1	98 ± 2	98 ± 1	98 ± 1	99 ± 1
**A + V**	99 ± 1	97 ± 2	96 ± 2	98 ± 1	98 ± 1	99 ± 1	99 ± 1	98 ± 1	98 ± 1	98 ± 2	99 ± 1	98 ± 2	99 ± 1

Time: As in Table 1. V = venom injection; A + V = venom injection followed by antivenom injection. Data presented as mean ± SD. No significant changes noted by two-way ANOVA.

## Data Availability

All data generated in the conduct of these experiments are presented in the manuscript.

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
