# Peer review of "Novel Toxicodynamic Model of Subcutaneous Envenomation to Characterize Snake Venom Coagulopathies and Assess the Efficacy of Site-Directed Inorganic Antivenoms"

_ijms, 2023, doi:10.3390/ijms241813939_

Round 1

Reviewer 1 Report (Previous Reviewer 2)

My previous concerns have been considered and addressed.

Author Response

“My previous concerns have been considered and addressed.”

            I appreciate the reviewer’s comment.

Reviewer 2 Report (New Reviewer)

This is an exciting research paper.

However, a few suggestions are placed that need to be clarified or included to further improve the manuscript.

Comment 1: Introduction is too long, it should be concise.

Comment 2: Was there any adverse events noted with 20 mg/kg of CORM-2 ?

Comment 3: Why are there references in the conclusion ?

Comment 4: Why was TEG/TOTEM was chosen ? These are rarely available in peripheral hospitals while most of the snake bite occurs in remote areas. Were conventional teats of coagulations like PT, aPTT, Febrinogen also done?

Comment 5: Towards the end, please mention, what will be the future use of these data or how will these findings be clinically relevant?

needs to be improved

Author Response

“This is an exciting research paper.”

“However, a few suggestions are placed that need to be clarified or included to further improve the manuscript.”

            I appreciate the reviewer’s kind comment, and I hope that my responses will be satisfactory.

“Comment 1: Introduction is too long, it should be concise.”

            I appreciate the reviewer’s comment and have tried to remove excess verbiage. I have reduced the text to approximately 1.5 pages, with the balance illustrations. I am unable to reduce it more and maintain its integrity.

“Comment 2: Was there any adverse events noted with 20 mg/kg of CORM-2 ?”

            As I mentioned in the Results, the rabbits appeared clinically unchanged after envenomation, without or with the dose of CORM-2.  Gross appearance, heart rate and arterial oxygenation were not affected. I have added an additional statement to the Results section involving the Mojave rattlesnake venom and antivenom treatment.

“Comment 3: Why are there references in the conclusion ?”

            I am unsure as to what the reviewer is referring to. There are references in the Discussion section as I compare the new findings with older literature and not limitations of the study. My Conclusions section has no references.

            However, jumping ahead to the reviewer’s Comment 5, I now include a Conclusions section concerning the matters mentioned.

“Comment 4: Why was TEG/TOTEM was chosen ? These are rarely available in peripheral hospitals while most of the snake bite occurs in remote areas. Were conventional teats of coagulations like PT, aPTT, Febrinogen also done?”

            I used TEG for several reasons. First, it is a method I have refined and published with for over 20 years in several clinical and basic research settings. The method provides a great deal of information beyond standard endpoint assays such as PT, aPTT and fibrinogen concentration. If the reviewer were to literature search TEG and envenomation, he/she would now find dozens of articles written not just by me but several other groups globally – most published after 2016, making this a young field. I have used the technique to mechanistically define the major action of 60 venoms from snakes and Helodermatidae, and to document in vitro the efficacy of the Ru-based antivenoms. TEG can differentiate fibrinolytic, thrombin-like, thrombin generating, and antiplatelet effects of venoms, dependent on proteome. I did not use the tests mentioned by the reviewer for these reasons.

            Regarding availability in peripheral hospitals, it is likely that advanced techniques may not be available to assess clinical treatment in countries with poor economies anytime soon. However, TEG is being eyed as a technique to use clinically in many countries, and it is being actively pursued by the Arizona Poison Center here in Tucson, Arizona. Thus, while the best methods for diagnosis may not be available globally, the best methods are still the best methods and should be used whenever possible.

“Comment 5: Towards the end, please mention, what will be the future use of these data or how will these findings be clinically relevant?”

            I had partially addressed this question in section 5, Conclusions. I added an additional sentence to emphasize the potential for clinical relevance.

In summary, I appreciate these comments, and I hope to have improved the manuscript to the satisfaction of the reviewer.

Round 2

Reviewer 2 Report (New Reviewer)

Looks Good!

Nil

This manuscript is a resubmission of an earlier submission. The following is a list of the peer review reports and author responses from that submission.

Round 1

Reviewer 1 Report

the revised manuscript is more informative than the original, and most my concerns were answered. I agree partially to the authors' response, but I still believe the injection volume of CORM2 was much (usually the volume is 1ml for human), and there was no evidence to show these CORM2 was completely absorbed subcutaneously within 15mins. 

Author Response

“the revised manuscript is more informative than the original, and most my concerns were answered. I agree partially to the authors' response, but I still believe the injection volume of CORM2 was much (usually the volume is 1ml for human), and there was no evidence to show these CORM2 was completely absorbed subcutaneously within 15mins.”

            I really want to satisfy this reviewer’s concerns, but I am uncertain how anyone can prove that all the CORM-2 was absorbed beyond general inspection of the injections site. The area of injection became nearly as flat as before against the animal’s flank, indicative that the fluid had gone somewhere. That somewhere is generally the lymphatic system in the case of subcutaneous injections, with perhaps some venous absorption as well. These issues have already been discussed in the manuscript. Further, how does anyone provide evidence of adsorption beyond visual inspection without damaging the animal with skin biopsies, etc.? Injecting between 4-6 ml under the loose skin of a 2-3 kg rabbit is not particularly remarkable in veterinary circles, and the rabbits I experimented on did not seem bothered by the consequent absorption over the three hour period.

            I am also puzzled by the comment that “usually the volume is 1ml for human”. To data, there are no human trials concerning the subcutaneous administration of CORM-2 to humans by any route. Also, being a physician, I have personally administered far more than 1 ml of medications subcutaneously to patients for over three decades without complications. Thus, I am frustrated that I cannot address this concern by the reviewer concerning human subcutaneous fluid administrations.

            In conclusion, I am not certain how I could have proven to the reviewer that the antivenom administered was absorbed post hoc beyond noting the event as a general observation. The CORM-2 was placed subcutaneously, did not leak from the injection site, and the area of injection flattened as previously mentioned.

Reviewer 2 Report

The author presents and useful study of anesthetized rabbits as a model for in vivo studies of venom induced coagulopathy and antivenom efficacy. It is clearly written, presented in a straightforward and helpful way that is useful for the field. I have outline a few places for clarification to improve readability and clarity below, and also highlight that data availability requirements need to be addressed.

General/Major:

It would be useful background in the introduction to mention what, if anything, is known about the mechanistic basis of Ru-based antivenom, and how it might be expected to differ from the more commonly used antibody-based antivenoms. Potentially compare and contrast the different mechanistic bases of each antivenom, and use to further justify the interest of the reader in use of a Ru-based approach.

Line 119: It is not initially clear that any statistics were conducted on the vital sign data, since no significant differences were found. It would be useful to mention the test conducted here, since this is a “results-first-methods-later” format. Then the reader will know stats were done to determine significance.

Author should briefly address why the plasmatic coagulation and subsequent calculation of platelet contribution to clotting was only done for the C. scutulatus venom, but not looked at for the other two species.

Data availability: The author states that all data generated in the conduct of these experiments are presented in the manuscript. This is not true. Only the summarized data (means +- SE) are presented in tables and graphs. The raw data used to calculate these statistics need to be included as supplemental files, so that it possible to access, recreate, and resuse the raw data from each test.

Minor:

Line 119: Need to define SpO2 at first use

Line 137-138: Is there prior work that validates this method for platelet mediated strength? If so please cite it. If not, please justify further.

Author Response

“The author presents and useful study of anesthetized rabbits as a model for in vivo studies of venom induced coagulopathy and antivenom efficacy. It is clearly written, presented in a straightforward and helpful way that is useful for the field. I have outline a few places for clarification to improve readability and clarity below, and also highlight that data availability requirements need to be addressed.”

            I appreciate the kind comments of the reviewer, and I hope that my responses satisfy his concerns about my work.

General/Major:

“It would be useful background in the introduction to mention what, if anything, is known about the mechanistic basis of Ru-based antivenom, and how it might be expected to differ from the more commonly used antibody-based antivenoms. Potentially compare and contrast the different mechanistic bases of each antivenom, and use to further justify the interest of the reader in use of a Ru-based approach.”

            I appreciate the reviewer’s thoughts on this matter, and the information concerning the purported mechanism by which Ru acts as an antivenom are found in the discussion on page 12. The focus on model development and potential molecular interactions of the various venoms with the coagulation system and its determination with viscoelastic methods takes up nearly three pages in the Introduction. I had ambivalence concerning making this section much longer, and instead chose to present the aforementioned Ru associated mechanisms in Discussion. I have added a few more passages in Discussion to provide some “compare and contrast” as requested on page 12, but an extensive treatment of the subject is beyond the scope of this work.

“Line 119: It is not initially clear that any statistics were conducted on the vital sign data, since no significant differences were found. It would be useful to mention the test conducted here, since this is a “results-first-methods-later” format. Then the reader will know stats were done to determine significance.”

            The lack of significance was noted in the text. I have made the changes requested to improve clarity, specifically naming the method used. Further, I have done these modifications for each table.

“Author should briefly address why the plasmatic coagulation and subsequent calculation of platelet contribution to clotting was only done for the C. scutulatus venom, but not looked at for the other two species.”

            I appreciate this comment, and it was originally addressed on page 6 as follows:

            2.2. Effects of B. moojeni envenomation on whole blood coagulation and efficacy of antivenom.

            The first rabbit of this series was dosed with 3.0 mg/kg [21]. While the first blood sample collected before envenomation was separated into whole blood and plasma without any problem, the sample collected one hour after envenomation was observed to have the citrate anticoagulated whole blood sample contain some clotted material. Further, the plasma retrieved from the centrifuged sample was also solidified despite anticoagulation. It was apparent that this venom contained calcium-independent, procoagulant enzymes that were rapidly acting at this dose. To deal this this issue, it was decided to rapidly collect whole blood and place it into the thrombelastographic cups with immediate activation with tissue factor as noted in the previously described methods. The time of collection to the time of onset of analysis was routinely under one minute, which should have outcompeted the procoagulant venom enzymes and allow an assessment of tissue factor initiated coagulation with the remaining components of the rabbit’s blood.

            To further improve clarity, I added a sentence to this passage to emphasize that the calculations cannot be made. I also added a sentence in the following sections concerning the following procoagulant venom to clarify why the calculation cannot be made.

“Data availability: The author states that all data generated in the conduct of these experiments are presented in the manuscript. This is not true. Only the summarized data (means +- SE) are presented in tables and graphs. The raw data used to calculate these statistics need to be included as supplemental files, so that it possible to access, recreate, and resuse the raw data from each test.”

            I respectfully disagree. It is a scientific standard that a measure of central tendency (mean, median) and variance (SD, SE) are included when reporting results. It took all of the data to generate these quantitative descriptors of the characteristics of the groups. Further, I did present means, but I presented standard deviations (SD), not standard errors of the mean. SD provides the raw variance that is not affected by number of subjects, and I feel strongly about always using this information. Presenting data in table and figure format with statistics is to provide the readership with an interpretable form of the data – all the data – so that the hypothesis is clearly upheld or disproven in an unambiguous manner. All my previous works in this journal and many others have not required all the raw data be presented, and this is the case for the vast majority of authors in all MDPI journals. In summary, I am presenting all my data in the standard format required by all scientific journals, and I am not being untruthful.

            However, I more than happy to provide the raw data for “recreation or reusage” purposes. I would also like to thank the reviewer for asking for this information, as it reminded me that one rabbit in one group could not have its heart rate and arterial oxygen saturation assessed secondary to a monitor malfunction (the rabbit bit the cable in half when I was not looking!). This information is now included in the revised manuscript. All raw data are now included in the requested supplemental file in a format convenient for statistical analyses.

Minor:

“Line 119: Need to define SpO2 at first use”

            I appreciate the reviewer finding this oversight. The changes requested have been made.

“Line 137-138: Is there prior work that validates this method for platelet mediated strength? If so please cite it. If not, please justify further.”

             I appreciate this comment. This method has been used in several publications. I am one of the progenitors of this method. I include my earliest citation with rabbits along with another that uses TTG comparisons in humans that compares whole blood and plasma instead of using platelet inhibition. I have used this methodology for over 20 years.

Nielsen VG, Geary BT, Baird MS. Evaluation of the contribution of platelets to clot strength by thromboelastography in rabbits: the role of tissue factor and cytochalasin D. Anesth Analg. 2000 Jul;91(1):35-9.

            Brodsky MA, Machovec KA, Chambers BP, Nielsen VG. Platelet-mediated thrombolysis in patients with δ-storage pool deficiency: a thrombelastographic analysis. Blood Coagul Fibrinolysis. 2011 Oct;22(7):610-2.

            The citations appear in the Methods section.

Round 2

Reviewer 1 Report

Thanks for the reply. I understand the difficulties to prove the scientificity of injection volume and absorption time. No more comments for this manuscript. 

Author Response

I appreciate the reviewer's comments. I have no way to post hoc assess this matter, and it is beyond the scope of the present work. The review states there are no more comments to be made. I have nothing to address, so I am at a complete loss as to why my paper was rejected.